# Sensitivity and specificity of rapid hepatitis C antibody assays in freshly collected whole blood, plasma and serum samples: A multicentre prospective study

**Beatrice N. Vetter**[1]*, **Stefano Ongarello**[1], **Alexander Tyshkovskiy**[2,3], **Maia Alkhazashvili**[4], **Nazibrola Chitadze**[4], **Kimcheng Choun**[5], **An Sokkab**[5], **Anja De Weggheleire**[6], **Fien Vanroye**[7], **Elena Ivanova Reipold**[1]

1 Foundation for Innovative New Diagnostics (FIND), Geneva, Switzerland, 2 Division of Genetics, Department of Medicine, Brigham and Women's Hospital, Harvard Medical School, Boston, Massachusetts, United States of America, 3 Belozersky Institute of Physico-Chemical Biology, Moscow State University, Moscow, Russia, 4 National Center for Disease Control & Public Health/R. Lugar Center for Public Health Research, Tbilisi, Georgia, 5 Sihanouk Hospital Center of Hope, Phnom Penh, Cambodia, 6 Neglegted Tropical Diseases Unit, Institute of Tropical Medicine, Antwerp, Belgium, 7 Reference Laboratory HIV/STD, Institute of Tropical Medicine, Antwerp, Belgium

* beatrice.vetter@finddx.org

**Data Availability Statement:** Replication Data for: HC013_FIND_Sensitivity and specificity of rapid hepatitis C antibody assays in freshly collected

## Abstract

### Background

This study evaluated performance of two hepatitis C virus (HCV) rapid diagnostic tests (RDTs) performed by intended users in resource-limited settings.

### Methods

Testing was conducted at three facilities in two countries (Georgia, Cambodia) using matched fingerstick whole blood, plasma and serum samples. Investigational RDTs were compared with a composite reference standard (CRS) comprised of three laboratory tests, and a reference RDT.

### Results

In matched samples from 489 HCV positive and 967 HCV negative participants, specificity with both investigational RDTs was high using either reference method ($\geq$98.4% in all sample types). Sensitivity was lower in whole blood versus plasma and serum for both RDTs compared with the CRS (86.5–91.4% vs 97.5–98.0% and 97.3–97.1%) and reference RDT (93.6–97.8% vs 100% and 99.4%). Sensitivity improved when considering only samples with detectable HCV viral load.

### Conclusion

Sensitivity was highest in serum and plasma versus whole blood. The World Health Organization prequalification criterion ($\geq$98%) was narrowly missed by both RDTs in serum, and one in plasma, possibly due to the intended user factor. Performance in whole blood was

whole blood, plasma and serum samples: a multicentre prospective study was published in Harvard Dataverse: https://dataverse.harvard.edu/dataset.xhtml?persistentId=doi:10.7910/DVN/W88BHP.

**Funding:** This work was supported by Unitaid, HCV grant number UA_HCV01, provided to the Foundation for Innovative New Diagnostics. All work carried out in the context of this study by each author was funded through this grant. https://unitaid.org/ The funder reviewed the clinical study protocol prior to commencement of the study for accuracy and completeness.

**Competing interests:** The authors have declared that no competing interests exist.

considered adequate, given potential roles of HCV infection history, improved sensitivity with detectable viral load and performance similarities to the reference RDT.

## Introduction

World Health Organization (WHO) member states have committed to the elimination of viral hepatitis as a public health threat by 2030 [1]. Screening for hepatitis C virus (HCV), a pathogen that affects approximately 71 million people worldwide (2015 estimate), is critical to the success of these targets, especially as only an estimated 20% of infected people are aware of their HCV status [1]. According to WHO recommendations, screening should be performed through the detection of HCV-specific antibodies using a single quality-assured serological *in vitro* diagnostic test, which can be either a laboratory-based immunoassay or a rapid diagnostic test (RDT) [2]. A positive RDT test is followed by confirmatory testing for viraemic infection via detection of HCV viral load (VL) or core antigen [2].

Low- and middle-income countries (LMICs) have the highest burden of HCV, representing over 70% of the global total [3]. However, access to laboratory-based testing services in these settings is often limited by the absence of suitable equipment, stringent training requirements and sample or patient transportation challenges. RDTs, which can be used outside of the laboratory, are an attractive alternative due to their affordability, ease of use and feasibility of utilizing various sample types, including plasma, serum, fingerstick whole blood or oral fluid [2]. WHO prequalification status intents to indicate that an RDT is likely to have reliable performance in LMICs, as it requires the generation of performance data in LMICs in intended use settings by intended users, with at least a portion of these data generated using freshly collected samples [4]. However, of the many commercially available HCV RDTs, only four have obtained WHO prequalification status to date [5]. The scarcity of quality-assured RDTs is an important barrier to HCV screening in LMICs on a large scale [6].

A previous retrospective study evaluated the performance of 13 HCV RDTs in archived plasma samples [7]. In this study, the majority of RDTs exhibited performance in line with WHO criteria for selection of HCV diagnostics in samples from patients without human immunodeficiency virus [HIV] co-infection (sensitivity ≥98% and specificity of ≥97% in serum or plasma samples [8, 9]). Sensitivity was lower in samples from HIV infected participants compared with samples from HIV uninfected participants; interestingly, the majority of false negative HIV infected samples did not have detectable HCV VL/core antigen. However, the retrospective study was performed on archived samples by highly trained staff in evaluation laboratories, a setting that does not fully reflect the reality in which HCV RDTs are intended or likely to be used. In the field, HCV RDTs are most likely to be performed in primary care or screening facilities by staff with limited training, using whole blood by finger prick as the most common sample type. Data on RDT performance in whole blood is often limited or absent, particularly in comparison with matched samples of other types.

The objective of the current study was to evaluate the sensitivity and specificity of HCV RDTs in a real-world setting. Performance was assessed in fresh, matched whole blood, plasma and serum samples that were collected and tested in resource-limited settings by intended users, i.e. nurses and primary healthcare personnel.

## Methods

### Study design

This prospective, multicentre study (NCT04139941) assessed the performance of two HCV RDTs: the HCV-Ab Rapid test (Beijing Wantai Biological Pharmacy Enterprise Co., Ltd,

Beijing, China) and the First Response HCV card test (Premier Medical Corporation Ltd., Mumbai, India). Operational characteristics of these tests are shown in S1 Table. These RDTs were selected as they met WHO prequalification criteria in archived plasma samples in the previous study [7], and the manufacturers had demonstrated a commitment to seeking WHO prequalification status.

Testing was conducted at three primary healthcare facilities in two countries. These were: a general outpatient clinic at the Sihanouk Hospital Center of Hope (SHCH), a non-governmental hospital providing low-cost medical care in Phnom Penh, Cambodia; an HCV screening facility at the National Center for Disease Control and Public Health (NCDC) in Tbilisi, Georgia; and an opioid substitution treatment facility at the Centre for Mental Health and Prevention of Addiction (CMHPA), also in Tbilisi, Georgia.

RDTs were tested on three sample types: fingerstick whole blood, ethylenediaminetetraacetic acid (EDTA) plasma, and serum (matched samples), all collected and tested on the same day. Performance was compared with three WHO prequalified laboratory reference tests, of which two were enzyme immunoassays (EIAs; Murex Anti-HCV version 4.0, Fujirebio INNOTEST HCV Ab IV) and one was a line immunoassay (LIA; Fujirebio INNO-LIA HCV Score), using a previously described composite reference standard (CRS) that incorporated the results of all three reference tests [7]. The algorithm was based on WHO prequalification evaluation protocols, with the final decision being based on the LIA test result. A signal-to-cut-off ratio of ≥1 (based on the measured optical density) was used for the EIAs; interpretation of LIA results was performed according to manufacturer instructions. Performance of the two investigational RDTs was also compared with a reference RDT, the WHO prequalified SD Bioline HCV test (Abbott Laboratories, Lake Bluff, USA; operational characteristics shown in S1 Table).

Reference testing was conducted at diagnostic reference laboratories (R. Lugar Center for Public Health Research, Tbilisi, Georgia and Biobykhin Medical Analysis Laboratory, Phnom Penh, Cambodia) using plasma samples, collected and tested on fresh or non-frozen samples (stored at 4˚C) within seven days of sample collection, in accordance with manufacturer instructions for use. Confirmatory testing to obtain HCV VL and genotyping information was performed on fresh plasma samples. Tests used for determination of HCV VL were the Real-Time HCV viral load assay (Abbott Laboratories, Lake Bluff, USA; limit of detection [LOD] 12 IU/mL) in Georgia and the AccuPid HCV Real-time PCR Quantification Kit (Khoa Thuong Biotechnology, Ho Chi Minh City, Vietnam; LOD 21 IU/mL) in Cambodia. Testing was performed between July 2019 and December 2019. Ethics approval for this study was obtained from the Cambodian National Ethics Committee for Health Research and the Georgian National Center for Disease Control and Public Health Institutional Review Board. Written informed consent was obtained from all study participants.

### Participant recruitment

Participants providing samples were required to be aged ≥18 years, have no history of HCV treatment (past or present), and be willing to perform an HIV test. At SHCH in Cambodia, all individuals visiting the facility as outpatients were invited to participate in the study until the daily recruitment target (~10 participants/day) was met. At CMHPA and NCDC in Georgia, all individuals visiting the facility were invited to participate. Additionally, known HCV positive individuals from the site databases were contacted and invited to participate. Participant demographic and medical history information was collected, including age, HIV status, other medications and infections, and recent vaccinations. Counselling related to HCV test results was offered, and all participants received HCV confirmatory testing. Participants were

assigned to the HCV positive and HCV negative group based on the result of the composite reference standard. If positive, Cambodian participants were given free treatment; Georgian participants received treatment via the national HCV treatment programme. The HCV status of participants was not known to RDT testers.

## RDT performance assessments

Every sample type was tested and interpreted once per RDT. Invalid results were repeated once, and plasma and serum samples were repeated in duplicate if the initial result was different to the reference RDT SD Bioline. Two lots of each RDT were used; the complete sample population was tested to approximately 50% with lot 1 and 50% with lot 2. Testers were nurses and primary healthcare personnel who are intended to perform RDT screening as per each countries' healthcare system. A number of different testers performed the tests at each site. The number of different testers for whole blood samples was 6, 2 and 4 at SHCH, CMHPA and NCDC, respectively. The corresponding numbers of testers for plasma and serum were 3, 2 and 3.

## Data capture

Participant demographic, medical history, RDT and LIA results were initially captured on paper case report forms. Viral load and genotype results, as well as EIA results were captured in electronic format on the respective analyses platforms. All data were subsequently entered into the electronic databased Open Clinica v4.0.

## Outcome measures

The primary outcome was the estimates of sensitivity and specificity of the two RDTs in each of the three sample types, compared with the CRS. Sensitivity and specificity compared with the reference RDT SD Bioline was a secondary outcome. For both outcomes, sensitivity and specificity were calculated for the overall sample set, by country and in the subset of samples with detectable HCV VL. Furthermore, statistical difference in performance between the sample types was assessed for both outcomes. Additionally, a multivariate analysis was performed to evaluate the impact of different demographic factors on RDT sensitivity in whole blood.

## Statistical analyses

For sample size calculations, sensitivity and specificity was assumed to be 90% for whole blood and 95% for plasma and serum samples. However, using these assumptions, the minimum sample sizes to achieve 80% power with a 95% CI of ±5% were lower than WHO Technical Specification Series-7 (TSS-7) requirements of 400 HCV positive and 1000 HCV antibody and RNA negative samples for diagnostic assessments of HCV RDTs [4]. Therefore, the TSS-7 values were used, with a 10% increase to account for sample exclusion due to indeterminate HCV status with the CRS (based on experience from the previous study [7]). Final sample size targets were 440 HCV antibody positive (HCV positive) and 1,100 HCV antibody negative (HCV negative) samples.

Point estimates with 95% confidence intervals based on Wilson's score method, were calculated for sensitivity and specificity. A performance comparison was performed using Pearson's chi-square test with Bonferroni adjustment to estimate statistical differences in RDT performance between sample types and by sample type between the two countries. Statistical analysis was performed using R (version 3.6).

Covariates included in the multivariate logistic regression were age, gender, presence of detectable viral load, HCV genotype and country. The model was applied separately for each

of the two investigational RDTs and the two reference methods (CRS and reference RDT SD Bioline). Estimates of coefficients and p-values were calculated using glm function with binomial logit specification in R.

## Results

### Population and sample characteristics

Of 1,540 individuals recruited, 11 were excluded, thus 1,529 samples of each type were provided in total. Characteristics of the individuals who provided samples are shown in Table 1. Mean age ranged from 40.3 years at CMHPA to 51.8 years at SHCH. Of the 1,529 samples, 489 were HCV positive, 966 were HCV negative, and 74 were excluded due to indeterminate results on the CRS (Fig 1). The number of HCV positive individuals encountered at NCDC in Georgia was higher than expected, thus more HCV positive participants were recruited than was anticipated in the predefined site enrolment targets.

HCV VL was detectable in 63% of HCV positive samples. HCV genotype 1, 1a and 1b were the most common, followed by genotype 3 and genotype 6. However, there were no genotype 3 samples from Cambodia, and no genotype 6 samples from Georgia (Table 2).

### Sensitivity and specificity versus composite reference standard

When compared with the CRS, specificity in the overall sample set was high ($\geq$98.4% for both RDTs in all three sample types), with no differences observed across sample types (adjusted p = 1.0) (Table 3). Sensitivity was lower in whole blood for the HCV-Ab Rapid test (86.5%) and the First Response HCV card test (91.4%), versus plasma (97.5% and 98.0%, respectively, adjusted p<0.001) and serum (97.3% and 97.1%, adjusted p<0.001 for the HCV-Ab Rapid test and adjusted p = 0.005 for the First Response HCV card test). Sensitivity was higher in the subset of samples with detectable HCV VL (>95.4% for both RDTs) for all sample types compared with the overall sample set.

Sensitivity in whole blood was considerably lower in Cambodia than Georgia for both RDTs (76.6% vs 94.2% for the HCV-Ab Rapid test and 85.0% vs 96.4% for the First Response HCV card test; adjusted p<0.001; Table 4). The majority of whole blood false negative samples with detectable VL from Cambodia were of genotype 1b, while those from Georgia were found

**Table 1. Study population characteristics.**

| | SHCH Cambodia (N = 770) | CMHPA Georgia (N = 439) | NCDC Georgia (N = 320) |
|---|---|---|---|
| Male, n (%) | 269 (34.9) | 360 (82.0) | 153 (47.8) |
| Mean age, years (range) | 51.8 (±13.7) | 40.3 (±10.5) | 42.6 (±13.6) |
| HCV positive on CRS, n (%) | 214 (27.8) | 209 (47.6) | 66 (20.6) |
| HIV positive, n (%) | 4 (0.5) | 0 | 0 |
| On ARV, n (%) | 2 (0.3) | 0 | 0 |
| On other medication[a], n (%) | 375 (48.7) | 259 (59.0) | 49 (15.3) |
| Other infections[a,b], n (%) | 43 (5.6) | 13 (3.0) | 8 (2.5) |
| Recent vaccination[a,c], n (%) | 46 (6.0) | 18 (4.1) | 23 (7.2) |

[a]All self-reported

[b]Hepatitis B virus, syphilis, hepatitis A virus, hepatitis D virus, influenza, measles, tuberculosis

[c]In the past 12 months; includes vaccination against hepatitis B virus, influenza, tetanus, rabies, human papillomavirus, measles-mumps-rubella, yellow fever. ARV, antiretroviral therapy; CMHPA, Centre for Mental Health and Prevention of Addiction; CRS, composite reference standard; HIV, human immunodeficiency virus; NCDC, National Center for Disease Control and Public Health; SHCH, Sihanouk Hospital Center of Hope.

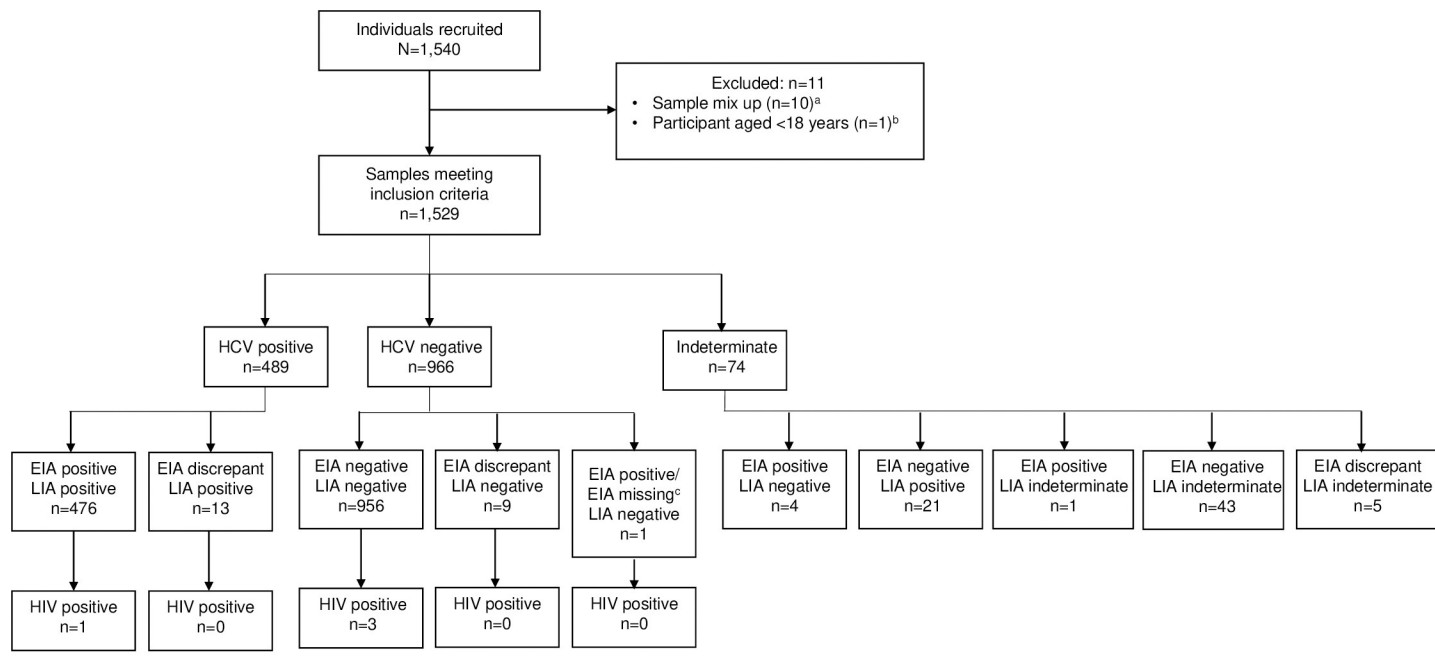

**Fig 1. Number of samples by HCV status.**

across all genotypes (S2 Table). No significant differences in sensitivity between the two countries were observed for plasma or serum, and no significant differences in specificity were observed between countries for any sample type (adjusted p>0.215).

Sensitivity and specificity of the reference RDT SD Bioline compared with the CRS are shown in S3 Table. Performance of this test was similar to the investigational RDTs in plasma

**Table 2. HCV VL and genotype status of HCV positive samples.**

|  | SHCH Cambodia | CMHPA Georgia | NCDC Georgia |
|---|---|---|---|
|  | (N = 214) | (N = 209) | (N = 66) |
| HCV VL status, n (%) |  |  |  |
| HCV VL undetectable | 79 (36.9) | 81 (38.8) | 23 (34.8) |
| HCV VL detectable | 135 (63.1) | 128 (61.2) | 43 (65.2) |
| Samples per HCV genotype, n (%) |  |  |  |
| 1, 1a, 1b | 63 (46.7) | 33 (25.8) | 22 (51.2) |
| 2 | 11 (8.1) | 11 (8.6) | 6 (14.0) |
| 3 | – | 60 (46.9) | 9 (20.9) |
| 4 | – | – | – |
| 5 | – | – | – |
| 6 | 59 (43.7) | – | – |
| Mixed | – | 20 (15.6) | 6 (14.0) |
| Not determinable | 2 (1.5) | 4 (3.1) | – |

SHCH, Sihanouk Hospital Center of Hope; CMHPA, Centre for Mental Health and Prevention of Addiction; NCDC, National Center for Disease Control and Public Health; HCV, hepatitis C virus; VL, viral load.

**Table 3. Investigational RDT performance versus composite reference standard in the overall sample set.**

| | TN, n | TP, n | FN, n | FP, n | Sensitivity, % (95% CI) | Specificity, % (95% CI) |
|---|---|---|---|---|---|---|
| | | | | Point estimates | | |
| **Whole blood (all samples)** | | | | | | |
| HCV-Ab Rapid | 958 | 423 | 66 | 8 | 86.5 (83.2, 89.2) | 99.2 (98.4, 99.6) |
| First Response HCV | 964 | 447 | 42 | 2 | 91.4 (88.6, 93.6) | 99.8 (99.2, 99.9) |
| **Whole blood (samples with detectable VL)** | | | | | | |
| HCV-Ab Rapid | — | 292 | 14 | — | 95.4 (92.5, 97.3) | — |
| First Response HCV | — | 301 | 5 | — | 98.4 (96.2, 99.3) | — |
| **Plasma (all samples)** | | | | | | |
| HCV-Ab Rapid | 951 | 477 | 12 | 15 | 97.5 (95.8, 98.6) | 98.4 (97.5, 99.1) |
| First Response HCV | 963 | 479 | 10 | 3 | 98.0 (96.3, 98.9) | 99.7 (99.1, 99.9) |
| **Plasma (samples with detectable VL)** | | | | | | |
| HCV-Ab Rapid | — | 304 | 2 | — | 99.3 (97.6, 99.8) | — |
| First Response HCV | — | 304 | 2 | — | 99.3 (97.6, 99.8) | — |
| **Serum (all samples)** | | | | | | |
| HCV-Ab Rapid | 955 | 476 | 13 | 11 | 97.3 (95.5, 98.4) | 98.9 (98.0, 99.4) |
| First Response HCV | 964 | 475 | 14 | 2 | 97.1 (95.3, 98.3) | 99.8 (99.2, 99.9) |
| **Serum (samples with detectable VL)** | | | | | | |
| HCV-Ab Rapid | — | 303 | 3 | — | 99.0 (97.2, 99.7) | — |
| First Response HCV | — | 303 | 3 | — | 99.0 (97.2, 99.7) | — |

| | Performance comparison (all samples), p-values | | | |
|---|---|---|---|---|
| | Sensitivity | | Specificity | |
| Sample type | HCV-Ab Rapid | First Response HCV | HCV-Ab Rapid | First Response HCV |
| Whole blood vs plasma | <0.001 | <0.001 | 1.0 | 1.0 |
| Whole blood vs serum | <0.001 | 0.005 | 1.0 | 1.0 |
| Plasma vs serum | 1.0 | 1.0 | 1.0 | 1.0 |

CI, confidence interval; FN, false negative; FP, false positive; TN, true negative; TP, true positive; VL, viral load

and serum (sensitivities of 95.1% and 93.9%, respectively), and even slightly lower in whole blood (90.4%). However, contrary to the other tests, no differences between sensitivity in whole blood samples and plasma or serum were observed for the reference RDT (adjusted p-value >0.160 for sensitivity and specificity across different sample types).

## Sensitivity and specificity versus RDT reference SD Bioline

When the RDT SD Bioline was used as a reference for comparison, specificity in the overall sample set was high for both investigational RDTs in all three sample types (≥96.8%), with no differences observed across sample types (adjusted p>0.099) (Table 5). For both investigational RDTs, sensitivity in whole blood increased when using the SD Bioline RDT as a reference (93.6% for the HCV-Ab Rapid test and 97.8% for the First Response HCV card test) and further increased in samples with detectable HCV VL (97.3% and 99.3%, respectively). Sensitivity in plasma and serum was also slightly increased when the RDT SD Bioline was used as a reference to evaluate performance (>99.4% for both sample types and RDTs). Sensitivity was considerably lower for both RDTs in whole blood compared with plasma (adjusted p<0.001 for the HCV-Ab Rapid test and adjusted p = 0.060 for the First Response HCV card test), and for the HCV-Ab Rapid test in whole blood compared with serum (adjusted p<0.001).

**Table 4. Investigational RDT performance versus composite reference standard by country.**

| | TN, n | TP, n | FN, n | FP, n | Sensitivity, % (95% CI) | Specificity, % (95% CI) |
|---|---|---|---|---|---|---|
| | | | | Point estimates | | |
| **Cambodia: whole blood** | | | | | | |
| HCV-Ab Rapid | 510 | 164 | 50 | 5 | 76.6 (70.5, 81.8) | 99.0 (97.7, 99.6) |
| First Response HCV | 515 | 182 | 32 | 0 | 85.0 (79.7, 89.2) | 100 (99.3, 100) |
| **Georgia: whole blood** | | | | | | |
| HCV-Ab Rapid | 448 | 259 | 16 | 3 | 94.2 (90.8, 96.4) | 99.3 (98.1, 99.8) |
| First Response HCV | 449 | 265 | 10 | 2 | 96.4 (93.4, 98.0) | 99.6 (98.4, 99.9) |
| **Cambodia: plasma** | | | | | | |
| HCV-Ab Rapid | 503 | 210 | 4 | 12 | 98.1 (95.3, 99.3) | 97.7 (96.0, 98.7) |
| First Response HCV | 512 | 211 | 3 | 3 | 98.6 (96.0, 99.5) | 99.4 (98.3, 99.8) |
| **Georgia: plasma** | | | | | | |
| HCV-Ab Rapid | 448 | 267 | 8 | 3 | 97.1 (94.4, 98.5) | 99.3 (98.1, 99.8) |
| First Response HCV | 451 | 268 | 7 | 0 | 97.5 (94.8, 98.8) | 100 (99.2, 100) |
| **Cambodia: serum** | | | | | | |
| HCV-Ab Rapid | 505 | 208 | 6 | 10 | 97.2 (94.0, 98.7) | 98.1 (96.5, 98.9) |
| First Response HCV | 513 | 209 | 5 | 2 | 97.7 (94.6, 99.0) | 99.6 (98.6, 99.9) |
| **Georgia: serum** | | | | | | |
| HCV-Ab Rapid | 450 | 268 | 7 | 1 | 97.5 (94.8, 98.8) | 99.8 (98.8, 100) |
| First Response HCV | 451 | 266 | 9 | 0 | 96.7 (93.9, 98.3) | 100 (99.2, 100) |

| | Performance comparison, p-values | | | |
|---|---|---|---|---|
| | Sensitivity | | Specificity | |
| **Cambodia vs Georgia** | HCV-Ab Rapid | First Response HCV | HCV-Ab Rapid | First Response HCV |
| Whole blood | <0.001 | <0.001 | 1.0 | 1.0 |
| Plasma | 1.0 | 1.0 | 0.541 | 1.0 |
| Serum | 1.0 | 1.0 | 0.217 | 1.0 |

CI, confidence interval; FN, false negative; FP, false positive; TN, true negative; TP, true positive

RDT sensitivity in whole blood was lower in Cambodia than in Georgia for both tests (87.4% vs 97.8%, adjusted p<0.001 for the HCV-Ab Rapid test and 95.1% vs 99.6%, adjusted p = 0.022 for the First Response HCV card test; Table 6). For both RDTs, specificity was lower in Cambodia compared with Georgia in plasma (94.5% vs 99.4%, adjusted p<0.001 for the HCV-Ab Rapid test and 96.6% vs 99.6%, adjusted p = 0.006 for the First Response HCV card test) and serum (94.2% vs 99.6%, adjusted p<0.001 for the HCV-Ab Rapid test and 96.2% vs 99.8%, adjusted p<0.001 for the First Response HCV card test). There were no significant differences between study countries in specificity for whole blood for either test. The multivariable logistic regression analysis showed that country was the most significant covariate associated with sensitivity (S4 Table). Besides the country, only gender was associated with sensitivity (slightly higher in males). However, gender only passed the threshold of statistical significance in one case (HCV Ab Rapid compared with the CRS).

## Discussion

In this prospective study of RDT performance in freshly collected whole blood, plasma and serum samples, sensitivity of both the HCV-Ab Rapid test and the First Response HCV card test was high in plasma and serum, but lower in whole blood. The concentration of antibodies is likely to be lower in whole blood compared with plasma and serum, which could explain the

**Table 5. Investigational RDT performance versus reference RDT in the overall sample set.**

| | TN, n | TP, n | FN, n | FP, n | Sensitivity, % (95% CI) | Specificity, % (95% CI) |
|---|---|---|---|---|---|---|
| | | | | **Point estimates** | | |
| **Whole blood (all samples)** | | | | | | |
| HCV-Ab Rapid | 1063 | 422 | 29 | 15 | 93.6 (90.9, 95.5) | 98.6 (97.7, 99.2) |
| First Response HCV | 1064 | 441 | 10 | 14 | 97.8 (96.0, 98.8) | 98.7 (97.8, 99.2) |
| **Whole blood (samples with detectable VL)** | | | | | | |
| HCV-Ab Rapid | — | 293 | 8 | — | 97.3 (94.8, 98.6) | — |
| First Response HCV | — | 299 | 2 | — | 99.3 (97.6, 99.8) | — |
| **Plasma (all samples)** | | | | | | |
| HCV-Ab Rapid | 1023 | 472 | 0 | 34 | 100 (99.2, 100) | 96.8 (95.5, 97.7) |
| First Response HCV | 1036 | 472 | 0 | 21 | 100 (99.2, 100) | 98.0 (97.0, 98.7) |
| **Plasma (samples with detectable VL)** | | | | | | |
| HCV-Ab Rapid | — | 304 | 0 | — | 100 (98.8, 100) | — |
| First Response HCV | — | 304 | 0 | — | 100 (98.8, 100) | — |
| **Serum (all samples)** | | | | | | |
| HCV-Ab Rapid | 1026 | 465 | 3 | 35 | 99.4 (98.1, 99.8) | 96.7 (95.4, 97.6) |
| First Response HCV | 1038 | 465 | 3 | 23 | 99.4 (98.1, 99.8) | 97.8 (96.8, 98.6) |
| **Serum (samples with detectable VL)** | | | | | | |
| HCV-Ab Rapid | — | 302 | 1 | — | 99.7 (98.2, 100) | — |
| First Response HCV | — | 302 | 1 | — | 99.7 (98.2, 100) | — |

| | Performance comparison (all samples), p-values | | | |
|---|---|---|---|---|
| | **Sensitivity** | | **Specificity** | |
| **Sample type** | **HCV-Ab Rapid** | **First Response HCV** | **HCV-Ab Rapid** | **First Response HCV** |
| Whole blood vs plasma | <0.001 | 0.060 | 0.136 | 1.0 |
| Whole blood vs serum | <0.001 | 1.0 | 0.099 | 1.0 |
| Plasma vs serum | 1.0 | 1.0 | 1.0 | 1.0 |

CI, confidence interval; FN, false negative; FP, false positive; TN, true negative; TP, true positive; VL, viral load

lower sensitivity seen in this study. However, although variability in sensitivity of HCV RDTs in whole blood has been previously reported in some studies [10–13], those that directly compared performance to plasma and serum have reported similar sensitivities across sample types [14, 15]. Other aspects that may have affected sensitivity include the possibility that some patients participating in the study had cleared their HCV infections, as evidenced by the absence of detectable VL in around one third of samples, and the improved sensitivity in the subset of samples with detectable viral load. Other studies have noted declines in HCV antibody levels following treatment-induced or spontaneous HCV clearance [16, 17], and a recent study observed reduced sensitivity of an HCV RDT in subjects with treatment-induced clearance [18]. While this would have affected all three sample types, it may have had a larger impact on sensitivity in whole blood as antibody concentrations would have been closer to the lower LOD compared with plasma and serum. Notably, WHO prequalification criteria are specifically designed for evaluation of plasma samples; no guidance is provided on expected performance in whole blood [8]. Given the variability in sensitivity in whole blood with HCV RDTs seen in earlier studies, achieving lower but acceptable sensitivity in whole blood may be considered adequate performance for the two investigational RDTs evaluated here. Nevertheless, HCV screening programmes using these RDTs must take into account the potential for lower performance in whole blood in real-world versus laboratory settings, particularly given

**Table 6. Investigational RDT performance versus reference RDT by country.**

| | TN, n | TP, n | FN, n | FP, n | Sensitivity, % (95% CI) | Specificity, % (95% CI) |
|---|---|---|---|---|---|---|
| | | | | Point estimates | | |
| **Cambodia: whole blood** | | | | | | |
| HCV-Ab Rapid | 575 | 159 | 23 | 13 | 87.4 (8.18, 91.4) | 97.8 (96.3, 98.7) |
| First Response HCV | 576 | 173 | 9 | 12 | 95.1 (90.9, 97.4) | 98.0 (96.5, 98.8) |
| **Georgia: whole blood** | | | | | | |
| HCV-Ab Rapid | 488 | 263 | 6 | 2 | 97.8 (95.2, 99.0) | 99.6 (98.5, 99.9) |
| First Response HCV | 488 | 268 | 1 | 2 | 99.6 (97.9, 100) | 99.6 (98.5, 99.9) |
| **Cambodia: plasma** | | | | | | |
| HCV-Ab Rapid | 536 | 203 | 0 | 31 | 100 (98.1, 100) | 94.5 (92.3, 96.1) |
| First Response HCV | 548 | 203 | 0 | 19 | 100 (98.1, 100) | 96.6 (94.8, 97.8) |
| **Georgia: plasma** | | | | | | |
| HCV-Ab Rapid | 487 | 269 | 0 | 3 | 100 (98.6, 100) | 99.4 (98.2, 99.8) |
| First Response HCV | 488 | 269 | 0 | 2 | 100 (98.6, 100) | 99.6 (98.5, 99.9) |
| **Cambodia: serum** | | | | | | |
| HCV-Ab Rapid | 539 | 196 | 2 | 33 | 99.0 (96.4, 99.7) | 94.2 (92.0, 95.9) |
| First Response HCV | 550 | 197 | 1 | 22 | 99.5 (97.2, 100) | 96.2 (94.2, 97.4) |
| **Georgia: serum** | | | | | | |
| HCV-Ab Rapid | 487 | 269 | 1 | 2 | 99.6 (97.9, 100) | 99.6 (98.5, 99.9) |
| First Response HCV | 488 | 268 | 2 | 1 | 99.3 (97.3, 99.8) | 99.8 (98.9, 100) |

| | Performance comparison, p-values | | | |
|---|---|---|---|---|
| | Sensitivity | | Specificity | |
| **Cambodia vs Georgia** | HCV-Ab Rapid | First Response HCV | HCV-Ab Rapid | First Response HCV |
| Whole blood | <0.001 | 0.022 | 0.145 | 0.221 |
| Plasma | N/A | N/A | <0.001 | 0.006 |
| Serum | 1.0 | 1.0 | <0.001 | <0.001 |

CI, confidence interval; FN, false negative; FP, false positive; TN, true negative; TP, true positive

that testing of fingerstick blood in non-laboratory settings is likely to be a common usage of these tests.

In our previous study using archived plasma samples [7], sensitivity of the investigational RDTs met the WHO prequalification sensitivity criterion of ≥98% [8], when compared with the laboratory-based CRS. In the current study, this criterion was narrowly missed by both RDTs in serum, and one of two in plasma. Unlike the previous study, in this evaluation the RDTs were performed by nurses and primary healthcare personnel, to represent a real-world setting. As such, variability in conditions, such as low lighting when reading RDTs, and user factors such as differences in line interpretation for low positive samples where lines can be more difficult to identify, could have impacted test performance. Similar factors, as well as the added technical challenge of fingerstick blood collection, may also have been a contributing factor to the lower sensitivity in whole blood. The fact that specificity was high in all sample types and sensitivity was close to WHO prequalification criteria in plasma and serum samples, suggests that the RDTs perform well in real-world settings and are likely to be beneficial to HCV screening programmes.

Consistent with our previous study in archived plasma samples [7], in this analysis, false negatives mostly occurred in samples with undetectable HCV VL. However, in our previous

study this effect was more apparent in HCV and HIV coinfected samples [7]. As only four participants in the current study were HIV positive, the effect of HCV VL on test performance in this study was not linked to HIV. Other studies have reported similar observations of improved HCV RDT performance in samples with detectable VL [19]. HCV VL testing is used to confirm viraemic infection in people who test positive for HCV antibodies [2], thus these samples represent participants who had active HCV infections. Because the sensitivity of the investigational RDTs was higher in samples with detectable VL compared with the overall sample set for all sample types, this provides some reassurance in the feasibility of using these RDTs to detect HCV in the people in need of treatment.

RDT test performance in Cambodia was considerably lower than in Georgia, in terms of sensitivity in whole blood compared with either reference test (CRS or reference RDT) and in terms of specificity in serum and plasma compared with the RDT reference. Differences in specificity when compared to the RDT reference might be explained by the lower sensitivity of the SD Bioline RDT in serum and plasma samples from Cambodia, resulting in a higher number of apparent false positives for the investigational RDTs.

The reason for the lower sensitivity of the investigational RDTs in Cambodia is not clear.

Although the majority of false negative samples with detectable VL from Cambodia were of genotype 1b, while those from Georgia were found across all genotypes, different methodologies were used at the different sites to determine HCV genotype, so it is difficult to determine whether this represents a meaningful difference. A prozone effect, whereby the ability of antibodies to form immune complexes is impaired at high concentrations, may also have resulted in false negatives, as has been shown with other RDTs [20]. Alternatively, it is possible that HCV positive participants from Cambodia with undetectable HCV VL had lower antibody titres, as suggested by the fact that proportionally, there were more true positives in samples with undetectable VL from Georgia compared with Cambodia (87.5% versus 50.6% for the HCV-Ab Rapid test and 92.3% vs 63.3% for the First Response HCV card test). Historically, the HCV epidemic in Cambodia has been largely driven through past unsafe medical practices [21, 22], whereas Georgia has an ongoing HCV epidemic in injection drug users [23]. Additionally, one of the two centres in Georgia was an opioid substitution treatment facility, thus a high proportion of Georgian participants would have been injection drug users. This suggests a possibility that the between-country differences in sensitivity may be due to Cambodian participants having generally cleared infections longer ago, while more Georgian participants had ongoing infections. Previous studies have shown that HCV screening tests can provide discrepant results in people with waning antibodies [24]. However, it was not possible to test this hypothesis in this study, as it was not designed to recruit participants to represent the proportionate occurrence of ongoing and past infections. Further research is needed to better understand sensitivity differences across different population groups or HCV endemic areas.

It is interesting to note that the WHO prequalified RDT SD Bioline, used as a reference RDT in this study, also had lower than expected sensitivity in whole blood in the overall sample set (including samples with and without detectable VL) when compared with the laboratory-based CRS. The quality of SD Bioline is well established [25, 26], thus this further highlights how regional and demographic differences in population can impact on RDT performance, even with established RDTs, and demonstrates the generally lower sensitivity of RDTs compared with laboratory-based immunoassays as antibody screening tests.

Specificity was high in all sample types for both investigational RDTs when compared with the CRS, meeting the WHO prequalification specificity criterion of ≥97% for HCV serology RDTs in plasma or serum specimens [8]. Specificity also met this criterion when compared with a WHO prequalified reference RDT test, except for one of two tests in plasma and serum samples, for which specificity dropped just below the threshold.

A limitation of this study is the stringent CRS used, which led to 73 samples being excluded from the study. While this provides confidence in the accuracy of the characterisation of the samples used in the study, it is possible that inclusion of the excluded samples would have affected sensitivity and specificity estimates. Additionally, the number of testers was higher for whole blood than for plasma and serum at two out of the three study sites, which may have contributed to differences in performance across sample types. However, previous studies have suggested that provision of training substantially reduces user errors with RDTs [27]. Training was provided to all testers involved in this study, thus the impact of user variability is likely to have been minimal.

In summary, both investigational RDTs performed well in fresh plasma and serum samples. Although sensitivity in whole blood performance was lower, particularly in Cambodia, given the potential impact of variability in HCV infection history, population drivers, conditions and user factors, data from other studies evidencing variable performance in whole blood with quality assured tests, and the fact that performance was similar to that of the reference RDT, test performance can be considered adequate. Additionally, overall performance in whole blood for samples with detectable VL was high. Comparative studies in different sample types should be taken into consideration when selecting HCV RDTs for screening programmes, bearing in mind that whole blood performance in real-world settings may be different from expectations based on data generated in laboratory evaluations.

## Supporting information

**S1 Checklist. TREND statement checklist.**
(PDF)

**S1 Table. Investigational and reference RDT operational characteristics.**
(DOCX)

**S2 Table. HCV genotype of false negative whole blood samples by country.**
(DOCX)

**S3 Table. Performance of the reference RDT test SD Bioline compared with the composite reference standard.**
(DOCX)

**S4 Table. Multivariable logistic regression analysis for RDT performance in whole blood (p-values).**
(DOCX)

**S5 Table. Protocol deviations.**
(DOCX)

**S1 File. Clinical study protocol.**
(PDF)

## Acknowledgments

The authors express their gratitude to all those involved in the execution of the project. Medical writing assistance and editorial support, under the direction of the authors, was provided by Rachel Wright, PhD, funded by Foundation for Innovative New Diagnostics (FIND), according to Good Publication Practice guidelines.

Both investigational HCV RDTs under evaluation were provided free of charge by the respective manufacturers.

## Author Contributions

**Conceptualization:** Beatrice N. Vetter, Stefano Ongarello, Elena Ivanova Reipold.

**Data curation:** Beatrice N. Vetter, Stefano Ongarello, Nazibrola Chitadze.

**Formal analysis:** Stefano Ongarello, Alexander Tyshkovskiy.

**Investigation:** Maia Alkhazashvili, Nazibrola Chitadze, Kimcheng Choun, An Sokkab.

**Methodology:** Beatrice N. Vetter, Stefano Ongarello, Elena Ivanova Reipold.

**Project administration:** Beatrice N. Vetter, Fien Vanroye.

**Supervision:** Maia Alkhazashvili, Nazibrola Chitadze, Kimcheng Choun, An Sokkab, Anja De Weggheleire, Fien Vanroye.

**Validation:** Beatrice N. Vetter, Stefano Ongarello.

**Writing – original draft:** Beatrice N. Vetter.

**Writing – review & editing:** Beatrice N. Vetter, Stefano Ongarello, Alexander Tyshkovskiy, Maia Alkhazashvili, Nazibrola Chitadze, Kimcheng Choun, An Sokkab, Anja De Weggheleire, Fien Vanroye, Elena Ivanova Reipold.

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
