## [Decision Letter · Decision Letter 0]

23 Sep 2020

PONE-D-20-21354

Sensitivity and specificity of rapid hepatitis C antibody assays in freshly collected whole blood, plasma and serum samples: a multicentre prospective study

PLOS ONE

Dear Dr. Vetter,

Thank you for submitting your manuscript to PLOS ONE. After careful consideration, we feel that it has merit but does not fully meet PLOS ONE’s publication criteria as it currently stands. Therefore, we invite you to submit a revised version of the manuscript that addresses the points raised during the review process. The reviewers' comments can be found at the end of this email.

We look forward to receiving your revised manuscript.

Kind regards,

Mohamed Fouda Salama, Ph.D.

Academic Editor

PLOS ONE

Journal Requirements:

Reviewers' comments:

Reviewer's Responses to Questions

**Comments to the Author**

1. Is the manuscript technically sound, and do the data support the conclusions?

Reviewer #1: Partly

Reviewer #2: Yes

Reviewer #3: Yes

2. Has the statistical analysis been performed appropriately and rigorously? 

Reviewer #1: Yes

Reviewer #2: I Don't Know

Reviewer #3: Yes

3. Have the authors made all data underlying the findings in their manuscript fully available?

Reviewer #1: Yes

Reviewer #2: Yes

Reviewer #3: No

4. Is the manuscript presented in an intelligible fashion and written in standard English?

Reviewer #1: Yes

Reviewer #2: Yes

Reviewer #3: Yes

5. Review Comments to the Author

Reviewer #1: This is a well conducted study evaluating the accuracy of two hepatitis C virus (HCV) rapid diagnostic tests (RDTs) in matched whole blood, plasma and serum samples compared to a WHO composite reference standard in three hospitals in Cambodia and Georgia. Statistical analysis was generally well conducted using the even too conservative approach of Bonferroni correction for p-values. I have a few questions, suggestions and comments:

Main Points

1. Known HCV positive individuals from the site databases were contacted and invited to participate in the study. How many of these were confirmed HCV+ by CSR? It would be interesting to do a sensitivity analysis restricting to only people with unknown HCV-status as this is the typical target population for screening.

2. The number of testers for full blood was larger than for the other type of samples. Could this explain some of the discrepancy?

3. The whole analysis focusses on sensitivity and specificity of the RDTs. Besides random variation and variability due to the testers, these are expected to be fixed features of the tests regardless of the setting in which the test was used. On the other hand, because DAA provide cure in 99% of cases and are well tolerated, the most important thing for screening and for individuals is a high positive predictive value (PPV). Because prevalence of HCV was much lower in Cambodia as opposed to Georgia, I expect the PPV also to be much lower. It would be good to quantify this and give the probability of being HCV+ and viremic in people who are tested positive with these RDTs.

4. A number of potential reasons for the lower sensitivity of the tests in full blood and in Cambodia have been provided but are rather speculative (undetectable HCV-RNA, older infections, HCV subtype, number and expertise of the testers, etc.). Average age was much older in the Cambodia site and could be used as a proxy of duration of infection. Is it not possible to identify which of these factors are more likely to explain the discrepancy by formally modelling the false positive rate (e.g. by multivariable logistic regression analysis)?

5. Eleven participants have been excluded. It would be good to describe the reasons for exclusion. It is also unclear why 73 samples were excluded. If each participant provided 3 samples, why 73 and not 33?

6. How were the Bonferroni corrected p-values actually obtained? Bonferroni suggested to use a change of the threshold for significance from the original α to α/k where k is the number of tests performed. To my knowledge, a Bonferroni-corrected p-value as such does not exist. Maybe the authors used the Benjamini–Hochberg false discovery rate correction? It would be a better approach, as Bonferroni is known to be too conservative.

7. Prequalification WHO sensitivity criterion was narrowly missed by both tested RDTs in serum, and one of two in plasma. This should be mentioned in the Conclusions of the abstract and the tone of the whole paragraph consequently lowered.

Other Points

1. Table 1. I would flip row and columns like in Table 2 (country as column headers). I would also include number and proportion of HCV+ by CRS and add percentages for all variables (HIV, on ARV, etc.) and p-values for difference between countries.

2. I would add a decimal figure to whole p-values even for those fully compatible with the null hypothesis (p=1.0).

3. Lines 259-261. Given the variability in sensitivity in whole blood with HCV RDTs seen in earlier studies, achieving lower but acceptable sensitivity in whole blood may be considered adequate performance for the two investigational RDTs evaluated here. Sentence seems a little too bald? Are not RDTs most useful as tests to be done outside of the laboratory on fingerstick whole blood samples?

Reviewer #2: This is a well written manuscript addressing one of the important issues in hepatitis C diagnostics - the use of DBS for detection of anti-HCV. I would have much preferred for authors to include additional WHO-prequalified RDTs.

Reviewer #3: This work shows the results of the comparison of a comprehensive number of two rapid diagnostic tests (RDT) for hepatitis C antibodies in the real world setting on a relatively high number of samples, and compared to a reference RDT and to a composite reference standard cconsisting in gold-stadard serology tests performed in the same samples.

The paper is well-writen and the main conclusion is that the two evaluated RTDs perform well in the filed, perhaps with decreased sensitivity for whole-blood samples and for negative viremia samples.

SPECIFIC COMMENTS:

-In the abstract include the numbers for sensitivity and specificiy; and mention the performance with respect to the composite reference.

- P5, L104-106. I would recommend movind this sentences ti the first paragraph in the "study design"

- P5, L112: What were the methods used for VL determination? What was the LOD? Was VL performed in freshly frozen samples? Pelase, specify.

- P5, L117. Plese detail inclusion and exclusion criteria.

- P6, L167. Specify why samples were excluded, based in the inclusion/exclusion criteria.

- P11, L204-205. The majority of FN in whole blood in Cambodia were HCV-1b..This is intringuing, do you have any potential explanation? What was the viral load for these samples?

- P14, L245 ...mention test name / brand here.

- P15, L254 other studies have also shown decreased sensitivity of Ab-RTDs after treatment-induced clearance in the clinical setting (see i.e. Carvalho-Gomes, Plos One, 2020)

MINOR COMMENTS:

p3, L58 ...ease of use and feasibility to...

p4 L69-70 ...lower in samples from HCV infected patientes as compared samples from HI uninfected...//...majority of FN anti-HCV in samples from HIV infected patients did not...

P16, L281...Because the sensitivity...

P16, L283...feasibility of the RDTs...

P16, L293 HCV genotypes...

P17, L328 ...particularly...

6. PLOS authors have the option to publish the peer review history of their article (what does this mean?). If published, this will include your full peer review and any attached files.

Reviewer #1: No

Reviewer #2: No

Reviewer #3: No

---

## [Author Response · Author response to Decision Letter 0]

4 Nov 2020

Reviewer #1 

This is a well conducted study evaluating the accuracy of two hepatitis C virus (HCV) rapid diagnostic tests (RDTs) in matched whole blood, plasma and serum samples compared to a WHO composite reference standard in three hospitals in Cambodia and Georgia. Statistical analysis was generally well conducted using the even too conservative approach of Bonferroni correction for p-values. I have a few questions, suggestions and comments:

1. Known HCV positive individuals from the site databases were contacted and invited to participate in the study. How many of these were confirmed HCV+ by CSR? It would be interesting to do a sensitivity analysis restricting to only people with unknown HCV-status as this is the typical target population for screening. 

Response: Although we included known HCV positive people in our study, their HCV status was unknown to the testers performing the RDTs. It is therefore unlikely that HCV status could have impacted our results. Additionally, none of the participants in this study had received or were receiving HCV treatment, and so our population may be considered comparable to those with unknown HCV status in this respect. 

2. The number of testers for full blood was larger than for the other type of samples. Could this explain some of the discrepancy? 

Response: We cannot exclude the possibility that user variability may have led to differences across sample types. Studies with RDTs for other diseases have noted that accuracy is highly-user dependent, however, user errors with RDTs have generally been related to issues relating to interpretation of manufacturer instructions, and are improved with the provision of training (Rennie et al. Trans R Soc Trop Med Hyg 2007;01:9–18). In this study, training was provided to all testers, so we believe the impact of user variability is likely to have been minimal. We have now mentioned this as a limitation on page 19 of the manuscript.

3. The whole analysis focusses on sensitivity and specificity of the RDTs. Besides random variation and variability due to the testers, these are expected to be fixed features of the tests regardless of the setting in which the test was used. On the other hand, because DAA provide cure in 99% of cases and are well tolerated, the most important thing for screening and for individuals is a high positive predictive value (PPV). Because prevalence of HCV was much lower in Cambodia as opposed to Georgia, I expect the PPV also to be much lower. It would be good to quantify this and give the probability of being HCV+ and viraemic in people who are tested positive with these RDTs. 

Response: Unfortunately, the design of this study makes it unsuitable for PPV calculation. Because a proportion of known positive participants were contacted for study participation, the study population was therefore not representative of the true prevalence in either country, which would have needed “random” recruitment. Also, in Cambodia, word of the study spread, which may have artificially increased participation of positive participants. Both of these factors may artificially increase the PPV of a test. 

4. A number of potential reasons for the lower sensitivity of the tests in full blood and in Cambodia have been provided but are rather speculative (undetectable HCV-RNA, older infections, HCV subtype, number and expertise of the testers, etc.). Average age was much older in the Cambodia site and could be used as a proxy of duration of infection. Is it not possible to identify which of these factors are more likely to explain the discrepancy by formally modelling the false positive rate (e.g. by multivariable logistic regression analysis)? 

Response: As per the reviewers suggestion, we performed a multivariable logistic regression analysis of RDT performance in whole blood, taking into account country, gender, age, detectable viral load and the different genotypes. In all cases, the country was the most significant covariate associated with sensitivity. Besides the country, only gender was associated with sensitivity (sensitivity was slightly higher in males). However, gender only passed the p-value threshold of 0.05 in one case (HCV Ab Rapid compared with the composite reference standard). Therefore, the difference in sensitivity between Cambodia and Georgia cannot be explained by the difference in characteristics of patients involved in these trials (at least, using these covariates). A table with results of the multivariable logistic regression analysis has been added as a supplement and a description of the methods added to the main text. 

5. Eleven participants have been excluded. It would be good to describe the reasons for exclusion. 

Response: Brief reasons for exclusion of these 11 samples are shown in Figure 1 (10 sample mix-ups and one participant <18 years of age). Additional details are as follows: The participant that was younger than 18 years of age was only 16 years old, but this was only noted after enrolment, upon review of the participant’s demographics and the date of birth. The remaining 10 samples were excluded as there was a mix up of the serum and plasma samples collected from these participants. These 10 participants were all recruited on the same day at a single site and serum/plasma aliquots were prepared on the same day. The mix up of samples occurred during the procedure of aliquoting into pre-labelled tubes. We have added these additional details to the footnote of Figure 1. 

6. It is also unclear why 73 samples were excluded. If each participant provided 3 samples, why 73 and not 33? 

Response: This refers to the samples that had indeterminate results on the CRS, not to the 11 participants who were excluded. As described on page 6, the CRS was composed of two enzyme immunoassays and one line immunoassay. Samples that had conflicting results between the two CRS assay types, or that had indeterminate results on the line immunoassay, were excluded. Please see Figure 1 for further details. Note that the value of 73 stated in the text at line 170 is in error; the correct number is 74, as shown in Figure 1. The text has been amended accordingly. 

7. How were the Bonferroni corrected p-values actually obtained? Bonferroni suggested to use a change of the threshold for significance from the original α to α/k where k is the number of tests performed. To my knowledge, a Bonferroni-corrected p-value as such does not exist. Maybe the authors used the Benjamini–Hochberg false discovery rate correction? It would be a better approach, as Bonferroni is known to be too conservative. 

Response: Bonferroni adjusted p-values were calculated with a "p.adjust" function in R. Benjamini-Hochberg FDR correction is another valid method of correction for multiple hypotheses, which is, indeed, less conservative than Bonferroni. The idea behind our choice of Bonferroni was related to the conservative behaviour of this approach as we aimed to identify only the most reliable statistically significant results and filter out as many false positives as possible. In this case the probability of having at least 1 false positive is equal to 5%, while in case of Benjamini-Hochberg there are, on average, 5% false positives among all significant results. To check whether our general conclusions are affected by the method of correction, we calculated adjusted p-values using Benjamini-Hochberg approach. This method resulted in similar p-values and did not alter our conclusions.

8. Prequalification WHO sensitivity criterion was narrowly missed by both tested RDTs in serum, and one of two in plasma. This should be mentioned in the Conclusions of the abstract and the tone of the whole paragraph consequently lowered. 

Response: We have amended the conclusions of the abstract accordingly.

8. Table 1. I would flip row and columns like in Table 2 (country as column headers). I would also include number and proportion of HCV+ by CRS and add percentages for all variables (HIV, on ARV, etc.) and p-values for difference between countries. 

Response: We have amended Table 1 as suggested. 

9. I would add a decimal figure to whole p-values even for those fully compatible with the null hypothesis (p=1.0). 

Response: We have made this change as suggested.

10. Lines 259-261. Given the variability in sensitivity in whole blood with HCV RDTs seen in earlier studies, achieving lower but acceptable sensitivity in whole blood may be considered adequate performance for the two investigational RDTs evaluated here. Sentence seems a little too bald? Are not RDTs most useful as tests to be done outside of the laboratory on fingerstick whole blood samples?

Response: We agree and have added the following sentence to page 16 of our manuscript: “Nevertheless, HCV screening programmes using these RDTs must take into account the potential for lower performance in whole blood in real-world versus laboratory settings, particularly given that testing of fingerstick blood in non-laboratory settings is likely to be a common usage of these tests.”

Reviewer #2

This is a well written manuscript addressing one of the important issues in hepatitis C diagnostics - the use of DBS for detection of anti-HCV. I would have much preferred for authors to include additional WHO-prequalified RDTs. 

Response: The aim of this study was to assess the performance of RDTs that do not currently hold WHO prequalification status, but for which the manufacturers had demonstrated a commitment to seeking WHO prequalification. Although there are already four HCV RDTs with prequalification status, as we mention on page 4 of our manuscript, there is a need for additional prequalified RDTs to facilitate HCV screening on a large scale in low- and middle-income countries. We included the WHO prequalified RDT SD Bioline (Abbott Laboratories) in our study as a reference RDT, to allow for comparison of the investigational RDTs with an already prequalified test. 

The reviewer may be interested in another article that we have recently published, which assessed the performance of 13 different HCV RDTs in archived plasma samples, including the four WHO prequalified tests (Vetter et al. J Infect Dis 2020; doi: 10.1093/infdis/jiaa389).

Reviewer #3: 

This work shows the results of the comparison of a comprehensive number of two rapid diagnostic tests (RDT) for hepatitis C antibodies in the real world setting on a relatively high number of samples, and compared to a reference RDT and to a composite reference standard consisting in gold-standard serology tests performed in the same samples. The paper is well-written and the main conclusion is that the two evaluated RTDs perform well in the field, perhaps with decreased sensitivity for whole-blood samples and for negative viremia samples.

1. In the abstract include the numbers for sensitivity and specificity; and mention the performance with respect to the composite reference. 

Response: We have added this information to the abstract as requested.

2. P5, L104-106. I would recommend moving this sentences to the first paragraph in the "study design"

Response: We believe the reviewer is referring to the description of the reference RDT, SD Bioline, suggesting that this be placed above the description of the composite reference test. However, as the comparison with the composite reference test was the primary outcome of the study (as mentioned in the Outcome Measures section on page 8), we believe it is more appropriate to detail this first. Sensitivity and specificity compared with the reference RDT was a secondary outcome, as defined in the study protocol.

3. P5, L112: What were the methods used for VL determination? What was the LOD? Was VL performed in freshly frozen samples? Please, specify. 

Response: The tests used for determination of viral load were the RealTime HCV viral load assay (Abbott Laboratories) in Georgia, and the AccuPid HCV Real-time PCR Quantification Kit (Khoa Thuong Biotechnology) in Cambodia. Limits of detection for these two tests are 12 and 21 IU/mL, respectively. Viral load assessments were performed on fresh samples, as per protocol. This information has now been added to page 6 of the manuscript.

4. P5, L117. Please detail inclusion and exclusion criteria. 

Response: Key inclusion criteria are detailed on page 7 of the manuscript, where it is stated that participants providing samples were required to be aged ≥18 years and have no history of HCV treatment (past or present). We have added the additional criteria that participants must have been willing to perform an HIV test. Other inclusion criteria are standard for clinical trials (i.e. provision of consent and willingness to undergo study procedures) and are therefore not described. Participants could have had known or unknown HCV serology, and could have been already registered at the local site or could register for the first time when enrolling in the study. There were no exclusion criteria other than inability to provide consent.

5. P6, L167. Specify why samples were excluded, based in the inclusion/exclusion criteria. 

Response: Brief reasons for exclusion of these 11 samples are shown in Figure 1 (10 sample mix-ups and one participant <18 years of age). Additional details are as follows: The participant that was younger than 18 years of age was only 16 years old, but this was only noted after enrolment, upon review of the participant’s demographics and the date of birth. The remaining 10 samples were excluded as there was a mix up of the serum and plasma samples collected from these participants. These 10 participants were all recruited on the same day at a single site and serum/plasma aliquots were prepared on the same day. The mix up of samples occurred during the procedure of aliquoting into pre-labelled tubes. As the study protocol did not contain instructions on how to manage sample mix ups, it was decided to exclude these samples from the study. We have added these additional details to the footnote of Figure 1. 

6. P11, L204-205. The majority of FN in whole blood in Cambodia were HCV-1b. This is intriguing, do you have any potential explanation? What was the viral load for these samples? 

Response: For the reviewer’s information, the viral loads (copies/mL) of the false negative samples with genotype information are shown in the response document. The samples from Cambodia have high viral loads, however, this is not surprising as samples with low viral load would have had insufficient genetic material to determine the genotype. While it is possible that genotype 1b may be more difficult to detect using these RDTs compared with other genotypes, we do not have supporting evidence for this given that the vast majority of genotyped samples from Cambodia were genotype 1b. The genotyping method used in Georgia was different to that used in Cambodia, thus results across countries cannot be compared.

7. P14, L245 ...mention test name / brand here. 

Response: Test names have been added as suggested (now page 15, line 259).

8. P15, L254 other studies have also shown decreased sensitivity of Ab-RTDs after treatment-induced clearance in the clinical setting (see i.e. Carvalho-Gomes, Plos One, 2020) 

Response: We thank the reviewer for drawing our attention to this important paper. We have now mentioned this study on page 16 of our manuscript.

9. p3, L58 ...ease of use and feasibility to...

Response: We have made the suggested edit (see page 4, line 62).

10. p4 L69-70 ...lower in samples from HCV infected patients as compared samples from HI uninfected...//...majority of FN anti-HCV in samples from HIV infected patients did not...

Response: We have made the suggested edit (see page 5, line 74).

11. P16, L281...Because the sensitivity...

Response: We have made the suggested edit (see page 17, line 299).

12. P16, L283...feasibility of the RDTs...

Response: We have made the suggested edit (see page 17, line 301).

13. P16, L293 HCV genotypes...

Response: We have made the suggested edit (see page 17, line 311).

14. P17, L328 ...particularly...

Response: We have made the suggested edit (see page 19, line 350).

---

## [Decision Letter · Decision Letter 1]

10 Nov 2020

PONE-D-20-21354R1

Sensitivity and specificity of rapid hepatitis C antibody assays in freshly collected whole blood, plasma and serum samples: a multicentre prospective study

PLOS ONE

Dear Dr. Vetter,

Thank you for submitting your manuscript to PLOS ONE. After careful consideration, we invite you to submit a revised version of the manuscript that addresses a final point raised by the reviewer regarding the data in one of the supplemental tables.

We look forward to receiving your revised manuscript.

Kind regards,

Mohamed Fouda Salama, Ph.D.

Academic Editor

PLOS ONE

Additional Editor Comments (if provided):

You need to address the reviewer's comment regarding data in S. table 4

Reviewers' comments:

Reviewer's Responses to Questions

**Comments to the Author**

1. If the authors have adequately addressed your comments raised in a previous round of review and you feel that this manuscript is now acceptable for publication, you may indicate that here to bypass the “Comments to the Author” section, enter your conflict of interest statement in the “Confidential to Editor” section, and submit your "Accept" recommendation.

Reviewer #1: (No Response)

2. Is the manuscript technically sound, and do the data support the conclusions?

Reviewer #1: Yes

3. Has the statistical analysis been performed appropriately and rigorously? 

Reviewer #1: Yes

4. Have the authors made all data underlying the findings in their manuscript fully available?

Reviewer #1: Yes

5. Is the manuscript presented in an intelligible fashion and written in standard English?

Reviewer #1: (No Response)

6. Review Comments to the Author

Reviewer #1: The authors addressed adequately all reviewers' query. My only last comment regards Supplemental Table 4. It would be useful to add the estimates of the OR and 95% CI and report the exact p-values rather than the cryptic '<0.05'

7. PLOS authors have the option to publish the peer review history of their article (what does this mean?). If published, this will include your full peer review and any attached files.

Reviewer #1: No

---

## [Author Response · Author response to Decision Letter 1]

13 Nov 2020

Reviewer #1

The authors addressed adequately all reviewers' query. My only last comment regards Supplemental Table 4. It would be useful to add the estimates of the OR and 95% CI and report the exact p-values rather than the cryptic '<0.05'

Response: Thank you for this comment, we have now added all p-values down to the third decimal point in Supplemental Table 4. While reviewing the table, we noted that the p-values in the second column for HCV-Ab Rapid compared to the reference RDT were incorrect. We have now corrected this, apologise for the error and note that all statements as made in the manuscript remain correct (lines 245-248).

---

## [Editor Report · Decision Letter 2]

16 Nov 2020

Sensitivity and specificity of rapid hepatitis C antibody assays in freshly collected whole blood, plasma and serum samples: a multicentre prospective study

PONE-D-20-21354R2

Dear Dr. Vetter,

We’re pleased to inform you that your manuscript has been judged scientifically suitable for publication and will be formally accepted for publication once it meets all outstanding technical requirements.

Kind regards,

Mohamed Fouda Salama, Ph.D.

Academic Editor

PLOS ONE
---

## [Editor Report · Acceptance letter]

26 Nov 2020

PONE-D-20-21354R2 

Sensitivity and specificity of rapid hepatitis C antibody assays in freshly collected whole blood, plasma and serum samples: a multicentre prospective study 

Dear Dr. Vetter:

I'm pleased to inform you that your manuscript has been deemed suitable for publication in PLOS ONE. Congratulations! Your manuscript is now with our production department. 

Kind regards, 

on behalf of

Dr. Mohamed Fouda Salama 

Academic Editor

PLOS ONE